# What is the impact of long-term COVID-19 on workers in healthcare settings? A rapid systematic review of current evidence

Moira Cruickshank[1]*, Miriam Brazzelli[1], Paul Manson[1], Nicola Torrance[2], Aileen Grant[2]

1 Health Services Research Unit, University of Aberdeen, Aberdeen, United Kingdom, 2 School of Nursing, Midwifery & Paramedic Practice, Robert Gordon University, Aberdeen, United Kingdom

* mcruickshank@abdn.ac.uk

## Abstract

### Background

Long COVID is a devastating, long-term, debilitating illness which disproportionately affects healthcare workers, due to the nature of their work. There is currently limited evidence specific to healthcare workers about the experience of living with Long COVID, or its prevalence, pattern of recovery or impact on healthcare.

### Objective

Our objective was to assess the effects of Long COVID among healthcare workers and its impact on health status, working lives, personal circumstances, and use of health service resources.

### Methods

We conducted a systematic rapid review according to current methodological standards and reported it in adherence to the PRISMA 2020 and ENTREQ statements.

### Results

We searched relevant electronic databases and identified 3770 articles of which two studies providing qualitative evidence and 28 survey studies providing quantitative evidence were eligible. Thematic analysis of the two qualitative studies identified five themes: uncertainty about symptoms, difficulty accessing services, importance of being listened to and supported, patient versus professional identity and suggestions to improve communication and services for people with Long COVID. Common long-term symptoms in the survey studies included fatigue, headache, loss of taste and/or smell, breathlessness, dyspnoea, difficulty concentrating, depression and anxiety.

**Data Availability Statement:** The data underlying the results presented in the study are available from the published studies included in the

systematic review. In addition, the relevant data are submitted as supporting material.

**Funding:** The authors received no specific funding for this work.

**Competing interests:** The authors have declared that no competing interests exist.

## Conclusion

Healthcare workers struggled with their dual identity (patient/doctor) and felt dismissed or not taken seriously by their doctors. Our findings are in line with those in the literature showing that there are barriers to healthcare professionals accessing healthcare and highlighting the challenges of receiving care due to their professional role. A more representative approach in Long COVID research is needed to reflect the diverse nature of healthcare staff and their occupations. This rapid review was conducted using robust methods with the codicil that the pace of research into Long COVID may mean relevant evidence was not identified.

## Introduction

Long COVID (LC) has rapidly emerged as a long-term debilitating illness [1] described by the World Health Organisation (WHO) as "devastating" [2]. Health and social care workers have a higher prevalence of self-reported LC compared to other occupational groups [3]. For health care workers (HCW), this is likely to be due to an increased risk of exposure and their central role in caring for patients with COVID-19, especially early in the pandemic when little was known about the virus, and many months before a vaccine was introduced [4].

The current joint NICE, SIGN and RCGP guideline on the management of long-term effects of COVID-19 (NG188) provides the following definitions:

- Acute COVID-19: Up to 4 weeks

- Ongoing symptomatic COVID-19: From 4 weeks up to 12 weeks

- Post- COVID-19 syndrome: Continuing for more than 12 weeks and not explained by an alternative diagnosis [5]

The term 'Long COVID' encompasses ongoing symptomatic COVID-19 and post-COVID-19 syndrome definitions above (i.e., signs and symptoms from 4 weeks after acute COVID-19).

Long COVID is an emerging condition for which a clear treatment care pathway or management options have yet to be established. However, given NHS workers have been disproportionately affected by LC, NHS England has put support measures in place, including occupational health, mental health hubs and guidance for health professionals returning to work, and for managers of these staff [6]. The Scottish Government has pledged £10 million over three years for LC support, although not specifically directed to NHS workers.

At present, there is limited evidence about narratives and experiences of those living with LC and their abilities to self-manage its consequences. In addition, there is little information specific to healthcare workers on the prevalence of LC, its pattern of recovery and its impact on healthcare resources.

This rapid systematic review focuses on the experiences of those working in healthcare settings and with LC symptoms, the impact on self-reported health, professional working lives, personal circumstances, and use of health services.

## Methods

A systematic rapid review was conducted and reported in adherence to the PRISMA 2020 statement and the Enhancing Transparency in Reporting the Synthesis of Qualitative Research

Statement [7, 8]. The methods for this appraisal were pre-specified in a research protocol (PROSPERO database registration number: CRD42021288181; https://www.crd.york.ac.uk/PROSPERO/display_record.php?RecordID=288181)

## Eligibility criteria

Eligible studies were written in English, published from December 2019 in a peer-reviewed journal and assessed participants with LC. Initial searches were conducted in November 2021 and were updated in December 2022. Evidence was considered from studies of any design reporting the experiences and/or impact of LC symptoms in HCW and including working performance, personal circumstances, or use of healthcare resources in healthcare workers. Eligible studies reported a definition of LC, or the criteria used to identify participants with LC symptoms. Clinical and non-clinical staff were eligible for inclusion. Social care staff and staff working in care homes and other long-term care facilities were not eligible for inclusion. Opinions and commentaries were excluded.

Studies reporting quantitative data only were grouped for narrative synthesis. Studies reporting qualitative data were grouped under emerging narratives and themes.

## Information sources and search strategy

A highly sensitive search strategy was developed by an information specialist (PM). The search strategy included database index and free-text terms to encompass the two facets of the longer-term effects of COVID-19 and all categories of workers in healthcare settings. A range of clinical and social science databases was searched, including Medline, Embase, CINAHL, Web of Science, PsycInfo, and ASSIA. There was no restriction on language or study type at the search stage. Results were limited to those published from December 2019. Searches were all carried out in November 2021 and updated in December 2022. The reference lists of all studies selected for full-text appraisal were screened for additional studies. A sample Medline search strategy is presented in S1 Appendix.

## Study selection

One reviewer (MC) screened all titles and abstracts identified by the initial and updated literature searches and a second reviewer (MB) screened those selected for full-text screening. One reviewer (MC) screened all potentially eligible full-text reports and those considered eligible were checked by a second reviewer (MB). Studies selected for inclusion were cross-checked by two experts (AG, NT).

## Data collection, quality appraisal and data synthesis

One reviewer (MC) conducted data extraction and a second reviewer (MB) checked the data extracted by the first reviewer. A third reviewer (AG) independently extracted data and cross-checked with the data agreed by the first two reviewers. For the updated searches, four reviewers (MC, AG, NT, MB) conducted data extraction and all extracted data were cross-checked by one reviewer (MC). At all stages, disagreements were resolved by consensus.

The following information was recorded from each included study: research question and setting, objectives and methods, demographic characteristics of participants, definition of LC, symptoms of LC, self-reported information on health status, effects of LC on working life or personal circumstances, use of healthcare services resources, and interpretation of findings from studies' authors.

The methodological quality of the included studies was assessed by a single researcher (MC) using the Quality of Reporting Tool (QuaRT) [9] and double checked by a second researcher (MB).

A pragmatic approach was adopted for the analysis of the results of the identified studies. Three researchers (MC, MB, AG) examined the qualitative studies to identify the main prominent and recurrent themes, organised the findings under 'descriptive' thematic headings and produced a holistic interpretation.

## Results

### Study selection

The initial literature searches identified 2089 records which were screened for relevance. Of these, 56 were considered potentially relevant and selected for full-text assessment. A total of 14 papers reporting 12 primary studies met the inclusion criteria. The updated searches identified 1681 records. A further 18 studies met the inclusion criteria and were included in the review, giving a total of 30 studies published in 32 papers.

A PRISMA flow diagram detailing the study selection process is presented in Fig 1.

### Studies' characteristics

Of the 30 included studies, two provided qualitative evidence [10, 11] and the remaining 28 survey studies used quantitative methods for collecting data on persistent COVID-19 symptoms [12–39]. The two qualitative studies were both conducted in the UK and recruited 43 participants [10] and 13 participants [11], respectively, with more than half of participants being medical doctors or GPs. Median age was 40 years in one study [10] and the age of most of participants in the other study was between 30 and 39 years [11].

The research question was described in most studies, but the study design was justified in only a few studies. Participant selection and recruitment were reported adequately in most studies, as were data collection and analysis methods. Only one study failed to report all domains adequately [13] while four studies reported all domains adequately [10, 11, 14, 15]. In general, the quality of the identified studies was judged to be satisfactory.

### Results of qualitative studies

From the two included qualitative studies [10, 11], we identified five themes related to the experience of health workers with LC:

1. Uncertainty about symptoms

2. Difficulty accessing services

3. Importance of being listened to and supported

4. Patient versus professional identity

5. Suggestions to improve communication and services for people with LC.

Each of these themes is summarised below.

**Uncertainty about symptoms.**   Participants experienced and described unfamiliar, unpredictable, and fluctuating symptoms which did not fit their clinical knowledge.

As healthcare workers, participants were able to recognise their physical and mental symptoms but struggled to make sense of the nature and duration of these symptoms and they

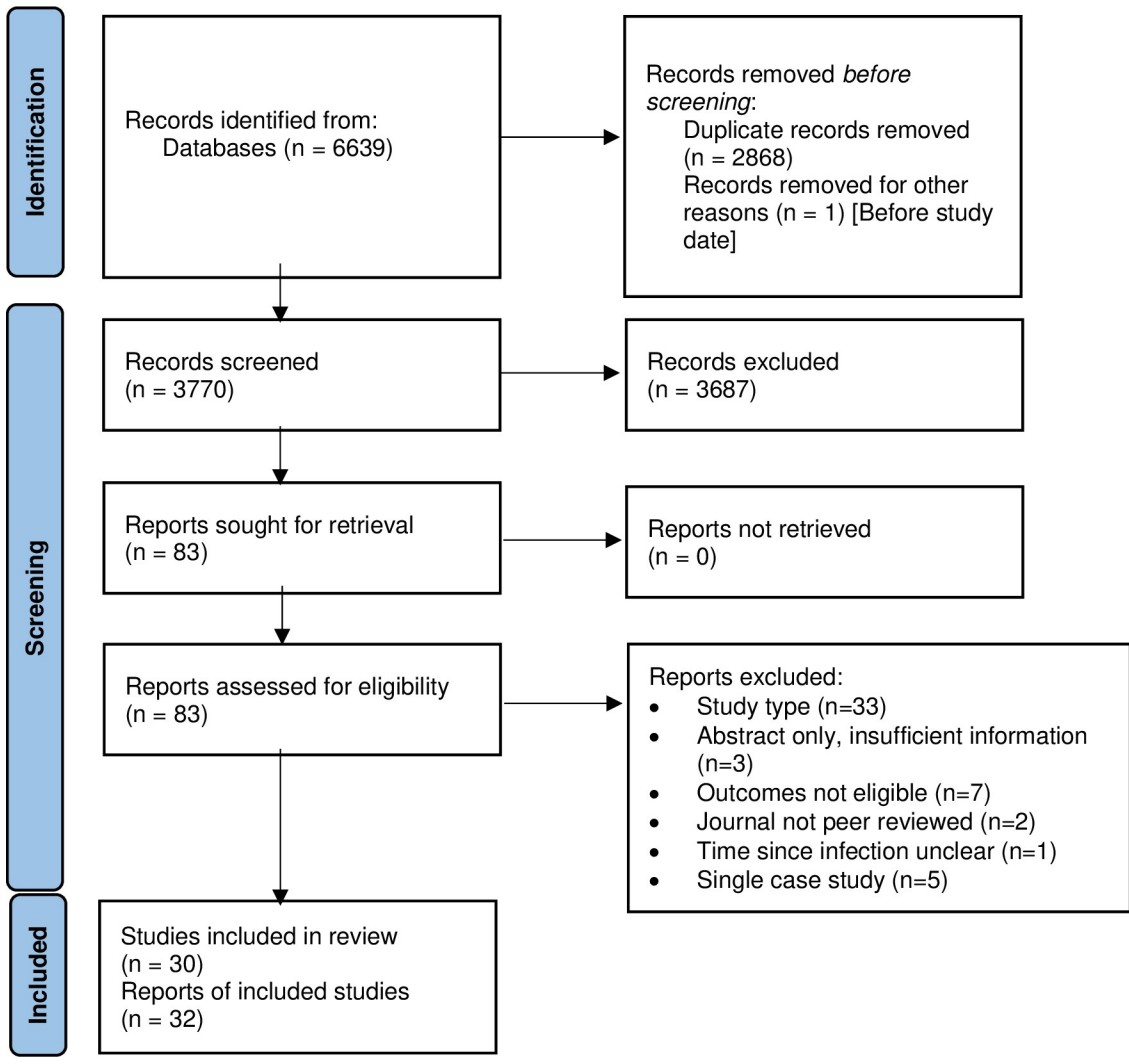

**Fig 1. PRISMA flow diagram.**

expressed concern about returning to work too soon or before the complete resolution of symptoms.

Most participants described a deterioration in their ability to carry out everyday tasks, including clinicians concerned about the safety of their practice, and raised concerns over whether they would ever recover or return to work.

> It's difficult because I keep getting new things, which is one of the frustrations of this. The brain stuff seems to be getting better, to the level that I can function. When the brain wasn't working that made me very scared because I need my brain! Not to be blasé, but with the chest pain and stuff I can still work because I can work remotely. If I don't have my brain I can't work, I can't plan, I can't string a sentence together. . . I did get a bit scared when I was ill for so long. . .(Doctor; Taylor 2021)[11]

Those who had returned to work worried that they were not able to cope at the required level or contribute enough to the workplace.

Participants turned to online social media groups for support and information. They also expressed newfound empathy for patients suffering from post-viral states and/or for those whose test results had not find anything conclusive.

It wasn't an active prejudice, but in the back of my mind I hadn't thought about it. . . a number of us in the group have said how ashamed we are of some of the attitudes we've had towards people, and lack of empathy. . . This concept of being irritated by patients when they're not really pleased when something comes back normal. . . Hopefully, it will make me a better and more empathetic doctor at the end. (Doctor; Taylor 2021) [11]

**Difficulty accessing services.** Participants described problems accessing and navigating care. They experienced delayed, absent, or inappropriate responses and perceived a lack of interest and support from their GPs in acknowledging and investigating their symptoms. In the study by Taylor et al., the doctors reported that their professional expertise had not been recognised or taken seriously, and some participants had called on personal contacts to secure appointments or referrals to specialists.

"I'd messaged a friend from medical school who's a cardiologist as I was wondering about pericarditis. . . I've always tried to be a good patient and go through my GP and things, but it wasn't working. So that's when I started messaging people and calling in favours." (Doctor; Taylor 2021) [11]

Participants reported also accessing private consultations for investigations, where positive test results helped them access specialist NHS referral.

*"My friend said "if you've got a mate in cardiology then ask for an echo". So I did. And I don't normally like to ask for favours. . . I reached out and he said "if you pay the fees for the echo then we'll do it". . . I felt disappointed I was unable to access this on the NHS. [. . .]" (Doctor; Taylor 2021) [11]*

Participants reflected on the lack of clinical pathways for LC and advocated a coordinated and multi-disciplinary approach.

**The importance of being listened to and supported.** Participants emphasised the value of being listened to by a clinician.

*"Then I spoke to my normal GP when she got back and that was probably the single most helpful conversation that I had during all of this because she, I was really struggling with how bad the fatigue was. . . I couldn't really have a shower without an hour's sleep afterwards and was feeling absolutely awful. Just feeling really grotty all the time. And she completely validated that I wasn't one of her nightmare patients." (Doctor; Taylor 2021) [11]*

Continuity of carer was important for participants as their story was often lengthy, unfamiliar, and multifaceted.

The focus when you do get a new GP speaking to you seems to be that they go back to the beginning and I've had a few consultations where I know I don't need to go to the hospital but your assessment is really all-around 'do I stay at home and wait this out or do I go to the hospital?' and there's nothing in between that. And I think if there was the same GP

who we are able to consult regularly they would build a picture of your baseline and I think that's what's lost with digital ways of working. (Doctor; Ladds 2020) [10]

Similarly, online LC support groups were considered important for reassurance, validation, and the opportunity to engage with others.

**Patient versus professional identity.** Combining their professional identity as healthcare workers with their role as patients was found particularly challenging by the participants.

I have found it very difficult to dissociate my doctor's brain from my patient's brain. I found it very difficult to. . . I'm a trainer as well, and I found it very difficult to dissociate my educator's brain from my patient's brain so I've had that dynamic going on for several weeks. I said to him "I hope I've handed over that locus of control, I'm putting trust in you, you're looking after me, I will go by your advice" (Doctor; Taylor 2021) [11]

Because of their own professional experience, participants were fully aware of the doctor-patient relationship and recognised that the uncertainty of their symptoms was somewhat difficult to address from a doctor's perspective. They feared they could have been perceived as a burden.

They were also frustrated by the fact that their doctors did not perceive and treat them as 'patients' and struggled to understand the expectation that, as healthcare workers, they were left to decide their own treatment.

[My GP] does rely heavily on me being a doctor and making my own management plan. . . There's a place for ICE [Ideas, Concerns and Expectations] but I need someone to be my doctor. If I don't come up with something, it's "wait and see", or a blood test (Doctor; Taylor 2021) [11]

Based on their experience of patients experiencing uncertain and persistent COVID symptoms, participants also reflected on their role as healthcare workers and their attitude in dealing with patients' concerns and requests in the past. Their own experience was an opportunity to re-evaluate the needs of patients and adopt a more sympathetic approach in the future.

**Suggestions to improve communication and services for people with Long COVID.** Based on their own experience of LC, participants felt an obligation to share their insight and raise awareness.

I mean not to sort of self-grandiose our group but there's a certain responsibility to put down our experiences so they can be opened up to other people who don't have the language and the access that we potentially have to communicate it to primary healthcare to access the services that need to be put in place for them (Doctor; Ladds 2021) [10]

Participants reflected on how to overcome the limitations of the current health services for patients with COVID. They advocated a multi-disciplinary approach to identify and address LC symptoms and the need for more personalised services.

My expectation of such a clinic would be to rule out treatable causes or complications, based on our symptoms. And then active involvement with physiotherapies and occupational therapies maybe a psychologist [. . .] we now know that COVID is a multi-system

disease so the fact that you don't display signs of respiratory infection doesn't mean that you don't have a problem. (Allied healthcare professional; Ladds 2021) [10]

Some participants also suggested establishing user-friendly online or telephone services to provide reliable information to people with LC.

### Results of quantitative studies

Twenty-eight of the included studies assessed the symptoms of healthcare workers experiencing LC. A summary of the characteristics of these studies is presented in Table 1, along with a summary of the studies' findings.

Overall, the proportions of healthcare workers experiencing long-term symptoms ranged from 23.1% at 6 months after infection [12] to 73% (in people with a positive nasopharyngeal swab; NPS; median 117 days since infection) [36]. The most common symptom was fatigue, [13–16, 18–27, 29, 30, 33, 35, 38, 39] with proportions ranging from 4% at 8 months after COVID-19 infection [14] to 75% at 1 month after COVID-19 infection [21]. Headache reports [14–17, 19–21, 24, 26, 29–31, 35, 39] ranged from 0.5% at > 60 days since infection [20] to 42% at 1 month after infection [21]. Proportions of loss of taste and/or smell [15, 17–20, 24–26, 28–31, 33, 35, 36, 39] ranged from 0.5% at up to 60 days and >60 days [20] to 51.1% at >3 months after infection [15]. Other commonly reported symptoms were dyspnoea [14–17, 21, 26, 28–32, 38], concentration/attention difficulties [14–16, 18, 20, 26, 29–32, 35], respiratory problems (i.e. breathlessness, breathing difficulties, shortness of breath) [13, 15, 18–20, 24, 35, 39] depression [16, 19, 21, 35, 37] and anxiety. [16, 19, 20, 27, 31, 35, 37]

Table 2 presents outcomes relating to working life, personal life, and healthcare use, all of which were scarcely reported by the included studies.

Six studies assessed the impact of LC symptoms on the working life of HCW. One study reported a median 10 missed working days in those with symptoms lasting <90 days and median 21 missed working days in people with symptoms lasting <365 days.[15] Four studies reported that workers' long-term symptoms disrupted their working life [14, 26, 28, 29]. Some participants reported that their social life and home life were disrupted by the persistence of their symptoms [14] and others reported being unable to participate in leisure activities because of their ongoing symptoms [18]. Conversely, one study reported that around three-quarters of HCW were leading a healthier lifestyle in the form of physical activity or taking multivitamins during the post COVID-19 recovery period [19].

### Risk of bias assessment

The findings of the Quality of Reporting Tool assessments for the 30 included studies are reported in Table 3.

### Discussion

Our systematic review identified 28 survey studies assessing the presence and impact of LC symptoms among HCW and two qualitative studies assessing their experiences and narratives. In general, quality assessment of the studies found them to be adequately reported.

HCW reported a wide range of diverse symptoms they have attributed to LC. The number and diversity of these LC symptoms have led to considerable challenges in achieving any formal diagnoses, investigations, management plans and prognosis, for those affected. This is reflected in the findings of this review.

Healthcare workers felt bewildered by symptoms and expressed dissatisfaction with access to the healthcare system, and the disengaged and dismissive attitude of some healthcare

**Table 1. Summary of characteristics of the 28 quantitative studies and summary of symptoms experienced by healthcare workers.**

| Study ID (country) | Sample size and participant characteristics | Healthcare workers experiencing long-term symptoms, n/N (%) | Details of symptoms, n (%) unless specified |
|---|---|---|---|
| **Brandt 2021** [12] **(Germany)** | Total n = 13<br>Occupation: NR<br>Age (range): 38.2 (23–55)<br>Sex: male 46.2%; female 53.8% | 3/13 (23.1) [6 months after infection] | • 1/13 (7.7%): Ongoing anosmia, ageusia and dysgeusia; fatigue after moderate physical activity<br>• 1/13 (7.7%): Unilateral paraesthesia in the ophthalmic nerve area<br>• 1/13 (7.7%): Intermittent weakness in one leg and numbness in the cheek |
| **Gaber 2021** [13] **(UK)** | Total n = 138<br>Occupation: NR<br>Age: NR<br>Sex: male 8.0%, female 92.0% | 61/138 (44.2) [time point NR] | • 54/138 (39.1%): moderate-to-severe fatigue<br>• 55/138 (39.9%): mild-to-moderate shortness of breath<br>• 67/138 (48.6%): sleep disturbance<br>• 61/138 (44.2%): mood disorders<br>• 42/138 (30.4%): struggling to cope with symptoms<br>• 3/11 (27.3%) males and 58/127 (45.7%) females: persistent symptoms |
| **Havervall 2021** [14] **(Sweden)** | Total n = 1395<br>Occupation: NR<br>Age, median (IQR):<br>• Seropositive: 43 (33–52)<br>• Seronegative: 47 (36–56)<br>Sex:<br>• Seropositive:<br>  ○ male 17.0%<br>  ○ female 83.0%<br>• Seronegative:<br>  ○ male 13.7%<br>  ○ female 86.3% | 84/323 (26.0)[a]<br>[≥2 months after infection]<br>48/323 (14.9)[a]<br>[≥8 months after infection] | • ≥1 symptom for ≥2mo: 84 (26.0%)<br>• ≥1 symptom for ≥8mo: 48 (14.9%)<br>• Symptoms lasting ≥2mo/8mo, n (%):<br>  • Anosmia: 47 (14.6%)/ 29 (9.0%)<br>  • Fatigue: 27 (8.4%)/ 13 (4.0%)<br>  • Ageusia: 25 (7.7%)/ 12 (3.7%)<br>  • Dyspnoea: 14 (4.3%)/ 6 (1.9%)<br>  • Sleeping disorder: 10 (3.1%)/ 7 (2.2%)<br>  • Headache: 9 (2.8%)/ 5 (1.5%)<br>  • Palpitations: 8 (2.5%)/ 2 (0.6%)<br>  • Concentration impairment: 7 (2.2%)/ 2 (0.6%)<br>  • Muscle/joint pain: 6 (1.9%)/ 2 (0.6%)<br>  • Memory impairment: 5 (1.5%)/ 1 (0.3%) |
| **Martinez 2021** [15] **(Switzerland)** | Total n = 260<br>Occupation: nursing staff 47.3%<br>Age: <30, 74 (28.5%); 30–49.99, 122 (46.9); > = 50, 64 (24.6%)<br>Sex: female 75.4%, male 24.6% | 69/260 (26.5%) [>3 months after infection] | 26.5% participants had not regained their usual level of health or had symptoms lasting >3 months. 45 participants reported details of symptoms.<br>• Fatigue: 68.9%<br>• Impaired taste or smell: 51.1%<br>• General weakness: 46.7%<br>• Concentration problems: 44.4%<br>• Breathing problems: 42.2%<br>• Sleep difficulties: 28.9%<br>• Headache: 22.2%<br>• Dizziness: 22.2%<br>• Chest pain: 20.0%<br>• Muscle pain: 20.0%<br>• Hair loss: 17.8%<br>• Palpitations: 15.6%<br>• Cough: 11.1%<br>• Joint pain: 8.9%<br>• Feverish feeling: 6.7%<br>• Decreased appetite: 6.7%<br>• Digestive problems: 4.4%<br>37 reported persisting symptoms over 365 days including 106 cumulative missed workdays; the most common reported symptoms among them were fatigue (5 participants, 100%), general weakness (4 participants, 80%) impaired sense of taste or smell and palpitations (3 participants, 60%).<br>32 patients who reported the diagnosis of SARS-CoV infection to have been made more than 365 days ago with a symptom duration of 365 days or less reported 303 cumulative missed workdays |

*(Continued)*

**Table 1.** (Continued)

| Study ID (country) | Sample size and participant characteristics | Healthcare workers experiencing long-term symptoms, n/N (%) | Details of symptoms, n (%) unless specified | |
|---|---|---|---|---|
| **Mattioli 2021 [16] (Italy)** | Total n = 120<br>Occupation: doctors & biologists 16.7%; nurses, physiotherapists & technicians 59.2%; health auxiliaries 24.2%<br>Age, years, median (range): 47.9 (26–65)<br>Sex: male 25.0%, female 75.0% | 78/120 (65%) [4 months after infection] | • Anosmia: 23 (19.2%)<br>• Fatigue: 18 (15%)<br>• Headache: 15 (12.5%)<br>• Attention difficulties: 14 (11.7%)<br>• Ageusia: 13 (10.8%)<br>• Dyspnoea: 13 (10.8%)<br>• Joint and muscle pain: 11 (9.2%)<br>• Insomnia: 8 (6.7%)<br>• Memory difficulties: 8 (6.7%)<br>• Irritability/anxiety: 6 (5%)<br>• Hair loss: 4 (3.3%)<br>• Arrhythmias: 3 (2.5%)<br>• Hearing loss: 2 (1.6%)<br>• Tremor: 2 (1.7%)<br>• Dizziness: 1 (0.8%)<br>• Radicular pain: 1 (0.8%)<br>• Cough: 1 (0.8%)<br>• Neurological deficits: 2 (1.7%)<br>• DASS-21 anxiety, median (range): 3 (0–18)<br>• DASS-21 stress, median (range): 7 (0–32)<br>• DASS-21 depression: 3 (0–30) | |
| **Nielsen 2021 [17] (Denmark)** | Total n = 840 (210 tested positive, 630 tested negative)<br>Occupation:<br>Positive test:<br>nursing staff 66.7%; medical doctors 18.1%; biomedical laboratory scientists 3.8%; medical secretaries 2.4%; other 9.0%<br>Negative test:<br>nursing staff 46.0%; medical doctors 17.6%; biomedical scientists 5.9%; medical secretaries 6.2%; other 24.3%<br>Age, n (%):<br>Positive test:<br><30y, 33 (15.7%); 30-39y: 49 (23.3%); 40-49y: 64 (30.5%); 50-59y: 49 (23.3%); > = 60y: 15 (7.1%)<br>Negative test:<br><30y, 58 (9.2%); 30-39y: 153 (24.3%); 40-49y: 221 (35.1%); 50-59y: 146 (23.2%); > = 60y: 52 (8.3%)<br>Sex:<br>Positive test: male 15.7%; female 84.3%<br>Negative test: male 15.7%; female 84.3% | Positive PCR/Negative PCR:<br>44.1%/20.2% [days 31–60][b]<br>38.5%/14.7% [days 61–90][b] | • Positive PCR, days 31-60/61-90, % of daily recordings[b]<br>• Any symptom: 44.1%/38.5%<br>• Reduced or lost taste or smell: 29.3%/28.6%<br>• Dyspnoea: 4.7%/3.5%<br>• Headache: 8.8%/6.6%<br>• Cough: 10.6%/4.1%<br>• Sore throat: 3.0%/2.8%<br>• Muscle ache or pain: 3.4%/3.6%<br>• Fever: 0.1%/0.0% | • Negative PCR, days 31-60/61-90, % of daily recordings[b]<br>• Any symptom: 44.1%/38.5%<br>• Reduced or lost taste or smell: 1.7%/0.9%<br>• Dyspnoea: 1.0%/0.5%<br>• Headache: 7.9%/5.3%<br>• Cough: 7.9%/5.5%<br>• Sore throat: 5.1%/4.0%<br>• Muscle ache or pain: 2.4%/2.3%<br>• Fever: 0.1%/0.1% |
| **Pereira 2021 [18] (UK)** | Total n = 38 (21 post-COVID-19 syndrome, 17 non-post-COVID-19 syndrome)<br>Occupation:<br>Post-COVID-19 syndrome: administrators 28.6%, dietician 4.8%, housekeeping 9.5%, physician 4.8%, nursing staff 28.6%, OT or physio 14.3%, pharmacists 9.5%<br>Non-post-COVID-19 syndrome: administrators 11.8%, housekeeping 5.9%, physicians 17.6%, nursing staff 35.3%, OT or physio 23.5%, phlebotomist 5.9%<br>Age, mean, years: Post-COVID-19 syndrome: 43; Non-post-COVID-19 syndrome 44<br>Sex, n (%):<br>Post-COVID-19 syndrome: male 4.8%, female 95.2%<br>Non-post-COVID-19 syndrome: male 29.4, female 70.6% | 21/38 (55.3) [7–8 months after symptom onset] | • Fatigue: 12/21 (57%)<br>• Loss of smell: 6/21 (29%)<br>• Breathlessness: 5/21 (24%)<br>• Difficulty concentrating: 5/21 (24%)<br>• 8/21 (38%) had 1 symptom; 6/21 (28.6%) had 2 symptoms and 7/21 (33.3%) had ≥3 symptoms<br>• Ongoing symptoms were more common in people of BAME origin (10/14) but the difference was NS<br>• Ongoing symptoms were more common in females (63%) than males (17%) but the difference was NS | |
| **Rao 2021 [19] (India)** | Total n = 163<br>Occupation: 51% doctors; 31% nurses; 9% AHP; 9% students<br>Age, years, %: <30: 52%, 30–40: 32%, 40–50: 11%, 50–60: 3%,<br>>60: 1%<br>Sex: 41% male, 59% female | NR | • Health issues in the post COVID-19 period:<br>• Fatigue on mild exertion: 42.9%<br>• Breathlessness: 8.6%<br>• Headaches, myalgia: 15.3%<br>• Fever, cough, sore throat: 5.5%<br>• Loss of taste, and smell: 21.5%<br>• Depression: 3.1%<br>• Anxiety: 6.1%<br>• None: 33.7% | |

*(Continued)*

**Table 1.** (Continued)

| Study ID (country) | Sample size and participant characteristics | Healthcare workers experiencing long-term symptoms, n/N (%) | Details of symptoms, n (%) unless specified |
|---|---|---|---|
| | | | • Frequency of health issues in the post COVID-19 period:<br>  • Almost daily: 28.2%<br>  • 3–4 times/week: 18.6%<br>  • Once a week: 9.7%<br>  • Occasionally, maybe once in 2 weeks: 43.6%<br>• Major concerns in the post COVID-19 period:<br>• Fear of contracting virus again: 46.5%<br>• Spreading to family members: 53.6%<br>• Developing post COVID-19 complications: 34.6%<br>  • Being isolated socially: 16.4%<br>  • Shortage of facilities: 5.7%<br>  • Financial: 17.0% |
| **Sultana 2021 [20] (Bangladesh)** | Total n = 186<br>Occupation: doctors 100%<br>Age, mean (SD), years:34.8 (9.9)<br>Sex: male 66.1%, female 33.9% | 44/186 (23.7)<br>[>60 days since infection] | • 44/186 (23.7%) reported at least one long post-COVID symptom (i.e. >60 days)<br>• 130/186 (69.9%) had at least one acute post-COVID symptom (up to 60 days)<br>• Symptoms 31–60 days/>60 days, n (%):<br>  • Difficulty breathing: 4 (2.2%)/12 (6.5%)<br>  • Cough: 2 (1.1%)/0<br>  • Palpitation: 2 (1.1%)/0<br>  • Chest pain: 1 (0.5%)/1 (0.5%)<br>  • Fatigue: 10 (5.4%)/15 (8.1%)<br>  • Sleep disturbance: 1 (0.5%)/7 (3.8%)<br>  • Lack of concentration: 3 (1.6%)/9 (4.8%)<br>  • Memory lapses: 1 (0.5%)/8 (4.3%)<br>  • Headache: 3 (1.6%)/1 (0.5%)<br>  • Anosmia: 4 (2.2%)/0<br>  • Irritability: 0/2 (1.1%)<br>  • Loss of taste: 1 (0.5%)/1 (0.5%)<br>  • Anxiety: 0/1 (0.5%)<br>  • Loss of appetite: 0/1 (0.5%)<br>  • Nausea: 1 (0.5%)/0<br>  • Joint pain: 0/3 (1.6%)<br>  • Hair fall: 0/8 (4.3%) |
| **Tawfik 2021 [21] (Egypt)** | Total n = 120<br>Occupation: Physicians, nurses, dentists and pharmacists (no further details reported)<br>Age, mean (SD), years: 33.7 (7.3)<br>Sex; male 42%, female 58% | NR | [c]Five most commonly reported symptoms at 1 month<br>• Fatigue: 75%<br>• Dyspnoea: 50%<br>• Depressive symptoms: 50%<br>• Headache: 42%<br>• Myalgia: 40%<br>[c]Five most commonly reported symptoms at 3 months<br>• Fatigue: 33%<br>• Dyspnoea: 29%<br>• Depressive symptoms: 20%<br>• Headache: 19%<br>• Bony aches: 18% |
| **Tempany 2021 [22] (Ireland)** | Total n = 217 (139 known infection, 78 assumed infections)<br>Occupation: NR<br>Age, range, years: 20–69<br>Sex, n (%):<br>Known infection: male 20.1%, female 79.9%<br>Assumed infection: male 19.2%, female 80.8% | 98/139 (70.5%)[d]<br>[≥12 weeks since infection] | 98/139 (70.5%) reported persistent symptoms, n (%):<br>• Fatigue: 78 (56.1%)<br>• Sleep disturbance: 56 (40.3%)<br>• Cognitive impairment: 34 (24.5%)<br>• Psychological symptoms: 30 (21.6%)<br>• Other physical symptoms: 30 (21.6%) |
| **Akova 2022 [23] (Turkey)** | Total n = 133<br>Occupation: physicians 33.8%, nurses/midwives 27.1%, other HCW 39.1%<br>Age, mean (SD), years: 36.0 (9.7)<br>Sex: male 45.1%, female 54.9% | 133/133 (100%) [HCW with Long COVID recruited] | 74/133 (55.6%) were fatigued/over-fatigued (i.e. Fatigue Assessment Scale score ≥22)<br>79/133 (59.4%) reported poor sleep quality (i.e. Pittsburgh Sleep Quality Index score ≥5) |

(Continued)

**Table 1.** (*Continued*)

| Study ID (country) | Sample size and participant characteristics | Healthcare workers experiencing long-term symptoms, n/N (%) | Details of symptoms, n (%) unless specified |
|---|---|---|---|
| **Carazo 2022 [24] (Canada)** | Total n = 6061 COVID-19 cases at 4 weeks, 1783 COVID-19 cases at 12 weeks and 4390 controls<br>Occupation: COVID-19 cases, physicians 4.0%, nurses 18.8%, nurse assistants 8.0%, healthcare assistants 25.9%, housekeepers 3.3%, administrators/ managers 10.0%, psychosocial workers 3.3%, other 26.5%<br>Age, mean (SD), years: hospitalised HCW 46.7 (11.9); non-hospitalised HCW 40.0 (12.1); controls 39.0 (10.4)<br>Sex: hospitalised HCW, male 29.7%, female 70.3%; non-hospitalised HCW, male 20.7%, female 79.3% | Non-hospitalised HCW: 46.2% with symptoms≥4 weeks, 39.9% ≥12 weeks<br>Hospitalised HCW: 76.3% with symptoms≥4 weeks, 67.6% ≥12 weeks | Hospitalised vs non-hospitalised HCWs with symptoms lasting ≥4 weeks:<br>• Fatigue: 30% vs 64%<br>• Loss of smell/taste: 20% vs 17%<br>• Shortness of breath: 20% vs 56%<br>• Cognitive dysfunction: 15% vs 33%<br>• Headache: 13% vs 23%<br>• Joint & muscular pain: 10% vs 22% |
| **Kameyama 2022 [25] (Japan)** | Total n = 83<br>Occupation: doctors 12.0%, nurses 62.7%, nursing assistants 8.4%, pharmacist 1.2%, technologists 9.3%, dental hygienists 3.6%, childcare worker 1.2%, clerk 1.2%<br>Age, median (IQR): 34.0 (25.0–48.0)<br>Sex: male 27.7%, female 72.3% | 60/83 (72.2%) at 1 month; 32/83 (38.6%) at 3 months; 17/83 (29.5%) at 6 months after infection | Most common symptoms at 1 month:<br>• Anosmia: 33.7%<br>• Fatigue: 33.7%<br>At 3 months:<br>• Anosmia: 18.1%<br>• Fatigue: 9.6%<br>At 6 months:<br>• Anosmia: 7.2%<br>• Fatigue: 4.8%<br>Median EQ-VAS score: 75.0<br>Median motivation for continuing to work score: 4 (0 = no motivation, 10 = maximum motivation) |
| **Kaplan 2022 [26] (Turkey)** | Total n = 121<br>Occupation: doctors 52.1%, nurses 24.8%, other 23.1%<br>Age, mean (SD): 33.5 (8.2)<br>Sex: male 32.2%, female 62.8% | 77/121 (63.6%) at >3 weeks after COVID-19 infection; 38/121 (31.4%) at >12 weeks after infection; 19/121 (24.6%) at >24 weeks after infection | Symptoms lasting > 3 weeks (n, %):<br>Fatigue (40, 33%), loss of smell (27, 22.3%), attention deficit/concentration disorder (25, 20.7%), dyspnoea (24, 19.8%), myalgia (24, 19.8%), loss of taste (23, 19%), cough (19, 15.7%), joint pain (18, 14.9%), sleep disturbance (14,11.6%), and memory difficulties (13, 10.7%)<br>Symptoms lasting >12 weeks (n, %):<br>Loss of smell (16, 13.2%), loss of taste (11, 9.1%), fatigue (10, 8.6%), attention deficit and concentration disorder (9, 7.4%), dyspnoea (8, 6.6%), sleep disturbance (7, 5.7%), cough (5, 4.1%), chest pain (4, 3.3%), memory difficulties (4, 3.3%), headache (3, 2.4%), myalgia (3, 2.4%), joint pain (1, 0.8%), sputum (1, 0.8%), constipation (1, 0.8%), and back pain (1, 0.8%)<br>Symptoms lasting >24 weeks (n, %):<br>Loss of smell (9, 11.6%), loss of taste (5, 6.4%), dyspnoea (5, 6.4%), headache (3, 3.8%), fatigue (2, 2.5%), cough (2, 2.5%), attention deficit and concentration disorder (2, 2.5%), memory difficulties (1, 1.2%), sleep disorder (1, 1.2%), back pain (1, 1.2%) |
| **Kinge 2022 [27] (South Africa)** | Total n = 62<br>Occupation: NR<br>Age, median (IQR): 33.5 (30–44)<br>Sex: male 24.2%, 75.8% female | 15/62 (24.2%) [symptoms experienced for ≥3 months] | • Persistent COVID-19 symptoms at three months and longer: 15 (24.2%)<br>• 33% of those with persistent symptoms reported more than one persistent symptom<br>Most commonly reported post-acute COVID-19 symptoms [timepoint NR]:<br>• Fatigue: 42%<br>• Anxiety: 34%<br>• Difficulty sleeping: 31%<br>• Chest pain: 24%<br>• Brain fog: 21%<br>• Muscle pain: 21%<br>• Joint pain: 18% |

(*Continued*)

**Table 1.** (Continued)

| Study ID (country) | Sample size and participant characteristics | Healthcare workers experiencing long-term symptoms, n/N (%) | Details of symptoms, n (%) unless specified |
|---|---|---|---|
| **Mendola 2022 [28] (Italy)** | Total n = 56<br>Occupation: physicians 33.9%, nurses 41.1%, nursing assistants 17.9%, other 3.6%<br>Age: median (IQR), 55 (50–61.2)<br>Sex: male 50.0%, female 50.0% | NR [questionnaire completed at mean 18 months since acute infection] | Post-COVID-19 symptoms among HCWs hospitalised due to COVID-19:<br>• Cough: 30 (57%)<br>• Resting dyspnoea: 33 (62%)<br>• Exertional dyspnoea: 46 (87%)<br>• Arthromyalgia: 38 (72%)<br>• Chest pain: 17 (32%)<br>• Tachycardia or palpitations: 19 (36%)<br>• Ageusia: 23 (43%)<br>• Anosmia: 25 (47%)<br>• Asthenia: 46 (87%)<br>• Cephalgia: 25 (47%)<br>• Loss of memory: 25 (47%)<br>• Hair loss: 22 (41%)<br>• Sleep disorders: 34 (64%)<br>• Anxiety/depression: 25 (47%) |
| **Mohr 2022 [29] (USA)** | Total n = 419<br>(Vaccinated, n = 180<br>Unvaccinated, n = 239)<br>Occupation: Non-clinical 30.5%, physicians 4.8%, advanced practice providers 2.9%, nurse/nurse assistants 39.1%, housekeeping 0.5%, other clinical 11.5%, other 10.7%<br>Age group, n (%): 18–29, 90 (21.5); 30–39, 167 (39.9); 40–49, 85 (20.3); 50–64, 77 (18.4)<br>Sex: male 15.3%, female 84.0%, non-binary 0.2%, missing data 0.5% | 298/419 (71%)<br>[6 weeks after illness onset] | [c]Prevalence of symptoms at 6 weeks after COVID-19 symptom onset<br>[Vaccinated (n = 180) / unvaccinated (n = 239):<br>• Fatigue: 35% / 48%<br>• Dyspnoea: 15% / 30%<br>• Cough: 25% / 30%<br>• Sinus congestion: 25% / 30%<br>• Myalgia: 15% / 25%<br>• Nausea: <5% / 10%<br>• Diarrhoea: 5% / 8%<br>• Sore throat: 8% /8%<br>• Chills: 0 / 5%<br>• Vomiting: 0 / 2%<br>• Fever: 0 / 1%<br>• Loss of taste or smell: 22% / 35%<br>• Headache: 20% / 30%<br>• Concentration problems: 25% / 25%<br>• Memory difficulties: 20% / 22%<br>• Dizziness: 10% /15%<br>• Confusion: 4% / 5%<br>• Movement disorders: <5% / <5%<br>• Trouble sleeping: 22% / 30%<br>• Exercise problems: 22% / 28%<br>• Chest pain: 6% / 10%<br>• Abdominal pain: <5% / <5% |
| **Nehme 2022 [30] (Switzerland)** | Total: n = 6639<br>(n = 3083 HCWs [65.0 tested negative, 35.0% positive] and n = 3556 general population)<br>Occupation all HCWs: 43.9% nursing staff, 19.3% administrative staff, and 15.9% physicians<br>Age, years, mean (SD):<br>All HCWs: 43.8 (11.0)<br>General population: 44.4 (14.4)<br>Sex:<br>All HCWs: female 72.3%, male 27.7%<br>General population: female 56.5%, male 43.5% | NR | Median time from infection to follow-up was 244 days (interquartile range IQR 202–400 days) in HCWs.<br>Presence of symptoms in COVID-19 positive HCWs vs negative HCWs, aOR (95%CI):<br>• Loss or change in smell: 11.79 (6.29, 22.09), p<0.001<br>• Loss or change in taste: 11.58 (5.23, 25.64), p<0.001<br>• Palpitations: 7.27 (2.09, 25.29), p = 0.002<br>• Dyspnoea: 3.71 (2.06, 6.70), p<0.001<br>• Difficulty concentrating/memory loss: 2.00 (1.30, 3.09), p = 0.002<br>• Fatigue: 1.59 (1.23, 2.06), p<0.001<br>• Headache: 1.60 (1.09, 2.34), p = 0.017<br>• Myalgia: 1.47 (0.92, 2.36), p = 0.109<br>• Arthralgia: 1.50 (0.92, 2.44), p = 0.102<br>• Cough: 1.60 (0.77, 3.32), p = 0.207<br>• Chest pain: 1.15 (0.36, 3.62), p = 0.811<br>• Exhaustion/burnout: 1.51 (0.92, 2.47), p = 0.100<br>• Insomnia: 1.26 (0.81, 1.97), p = 0.300<br>• Stress: 0.59 (0.30, 1.19), p = 0.141 |

*(Continued)*

**Table 1.** (Continued)

| Study ID (country) | Sample size and participant characteristics | Healthcare workers experiencing long-term symptoms, n/N (%) | Details of symptoms, n (%) unless specified |
|---|---|---|---|
| **El Otmani 2022** [31] (Morocco) | Total n = 118 infected with COVID-19, n = 118 not infected<br>Occupation: COVID-19 cases, doctors 78%, other 22%<br>Age, mean (range): infected, 29 (21–54), not infected, 29 (21–54)<br>Sex: infected, male 29%, female 71%; not infected, male 29%, female 71% | 56/118 (47.4%) [timepoint NR] | 56/118 (47.4%) experienced at least one symptom of Long COVID:<br>• Anosmia/hyposmia: 9.6%<br>• Dysgeusia: 6%<br>• Tinnitus: 7.2%<br>• Dyspnoea: 3.6%<br>• Cough: 4.8%<br>• Chest pain: 8.4%<br>• Palpitations: 10.8%<br>• Myalgia: 13.3%<br>• Arthralgia: 9.6%<br>• Abdominal pain: 4.8%<br>• Diarrhoea: 6%<br>• Itching: 1.2%<br>• Headache: 12%<br>• Dizziness: 8.4%<br>• Sensitive disorders: 1.2%<br>• Sleep disorders: 12%<br>• Anxiety: 21.7%<br>• Attention disorders, memory impairment, brain fog: 14.4% |
| **Pilmis 2022** [32] (France) | Total n = 74 included in the 7-month cohort<br>Occupation: NR<br>Age, median, years: 47 [IQR 33.2–54.2 years]<br>Sex: female 82.4%, male 17.6% | 24/74 (32.4%) [7-month cohort study] | • Asthenia: 12 (16.2%)<br>• Dyspnea: 10 (13.5%)<br>• Concentration disorder: 7 (9.5%) |
| **Selvaskandan 2022** [33] (UK) | Total n = 423 (120 with COVID-19 diagnosis)<br>Occupation: doctors 34%, nurses 36%, other multidisciplinary professionals 30%, retired (<1%)<br>Age group:<br><25 years, 2 (0.4%)<br>25–34 years, 63 (15%)<br>35–44 years, 110 (26%)<br>45–54 years, 151 (36%)<br>55–64 years, 91 (22%)<br>≥65 years, 5 (1%)<br>Sex: female 74%, male 26% | 43/120 (36%) [beyond 3 months after infection] | • Fatigue: 30 (70%)<br>• Mood changes: 8 (19%)<br>• Ageusia/anosmia: 6 (14%) |
| **Senjam 2022** [34] (India) | Total n = 395 hospital employees<br>Occupation: NR<br>Age: NR<br>Sex: NR | 156/395 (39.5%) | 39.5% of hospital employees reported post-COVID symptoms at ≥4 weeks after infection. The multivariable regression analysis showed that non-healthcare staff were at lower risk of having post-COVID symptoms than employees working in the hospital (OR: 0.65, 95%CI 0.74–3.87) |
| **Strahm 2022** [36] (Switzerland) | Total n = 3334 (556 with positive NPS, 228 only seropositive, 2550 negative controls)<br>Occupation: Positive NPS, nurses 59%, physicians 13%, other 25%; only seropositive, nurses 57%, physicians 11%, other 26%; negative controls, nurses 41%, physicians 17%, other 38%<br>Age, median (range): Positive NPS, 38.9 (16.8–63.5); only seropositive, 37.9 (17.1–63.9); negative controls, 41.0 (16.5–72.6)<br>Sex: Positive NPS, male 18%, female 81%; only seropositive, male 20%, female 80%; negative controls, male 21%, female 79% | Proportion of HCWs reporting one or more symptoms compatible with Long COVID:<br>• Positive nasopharyngeal swab (NPS): 73%<br>• Only seropositive: 58%<br>• Negative controls: 52% | The most common symptoms were exhaustion/burnout (33% in NPS-positive vs. 25% in only seropositive vs. 24% in negative controls) and weakness/tiredness (34% vs. 25% vs. 22%). Impaired taste/olfaction (33% vs. 16% vs. 6%) and hair loss (17% vs. 17% vs. 10%) were the only symptoms which were significantly more common in only seropositive HCW compared to negative controls |
| **Uvais 2022** [37] (India) | Total n = 107<br>Occupation: NR<br>Age group, n (%): 20–30, 84 (78.5); 31–40, 19 (17.8); 41–50, 4 (3.7)<br>Sex: female 63.6%, male 36.4% | 73/102 (71.6) [timepoint NR] | 73/102 (71.6%) reported persistent symptoms<br>• Depression: 26.2%<br>• Anxiety: 12.1%<br>• PTSD: 3.7% |

*(Continued)*

**Table 1.** (*Continued*)

| Study ID (country) | Sample size and participant characteristics | Healthcare workers experiencing long-term symptoms, n/N (%) | Details of symptoms, n (%) unless specified |
|---|---|---|---|
| **D'Avila 2023 [38] (Brazil)** | Total n = 289 (n = 174 at 6 weeks) Occupation: NR Age, mean (SD): 42.2 (9.5) Sex: male 19.4%, female 80.6% | 63/174 (36.2%) [6 months after acute COVID-19 infection] | 63/174 (36.2%) diagnosed with post-COVID-19 syndrome: <br>• Fatigue: 23/63 (36.5%) <br>• Sleep disturbance: 9/63 (14.3%) <br>• Dyspnoea: 8/63 (12.7%) <br>• Cough: 6/63 (9.5%) <br>• Reduced QoL due to post-COVID-19 syndrome: 63/85 (74.1%) |
| **Shukla 2023 [35] (India)** | Total n = 679 Occupation: doctors 39.8%, nurses 26.7%, paramedical and ancillary staff 33.6% Age group, n (%): <45 years, 596 (87.8%) >45 years, 83 (12.2%) Sex: female 50.8%, male 49.2% | 206/679 (30.3%) [between 12–52 weeks after COVID infection] | • Fatigue: 78 (11.5%) <br>• Pain in joints: 34 (5%) <br>• Soreness in muscles: 30 (4.4%) <br>• Fever: 19 (2.8%) <br>• Difficulty in breathing during physical activity: 41 (6%) <br>• Cough: 31 (4.6%) <br>• Tightness in chest: 15 (2.2%) <br>• Throat Pain: 14 (2%) <br>• Difficulty in breathing while at rest: 11 (1.6%) <br>• Sensation of irregular or fast heartbeat: 8 (1.2%) <br>• Reduced Appetite: 9 (1.3%) <br>• Nausea: 7 (1%) <br>• Diarrhoea: 6 (0.9%) <br>• Abdominal pain: 4 (0.6%) <br>• Sore throat: 13 (2%) <br>• Pain in the ear: 3 (0.4%) <br>• Ringing sensation in ears: 2 (0.3%) <br>• Headache: 31 (4.6%) <br>• Loss of smell: 31 (4.6%) <br>• Loss of taste: 27 (4%) <br>• Difficulty in concentrating: 14 (2%) <br>• Difficulty to focus on the usual things: 12 (1.8%) <br>• Forgetting things easily: 11 (1.6%) <br>• Pins & needles sensation or numbness in hands or feet: 7 (1%) <br>• Difficulty in thinking clearly or getting anything done: 5 (0.7%) <br>• Sleep disorder (Insomnia): 58 (8.5%) <br>• Depression: 9 (1.3%) <br>• Stress: 7 (1%) <br>• Anxiety: 1 (0.2%) <br>• Skin rash: 7 (1%) <br>COVID sequelae were significantly higher among HCWs $\geq$ 45 years of age (OR 2.03; 95% CI 1.27–3.25) and those with comorbidity (OR 2.01). In contrast, the odds of having sequelae were found to be significantly lesser among males (OR 0.55) and among doctors as well as doctors and nursing staff as a combined group compared to other HCWs (OR 0.65 and 0.70, respectively). Logistic regression analyses confirmed that moderate-severe COVID was an independent predictor for risk of having COVID sequelae (adjusted OR 5.83; 95% CI 3.05–11.14) and male gender was a protective factor (adjusted OR 0.56; 95% CI 0.4–0.8) |

(*Continued*)

**Table 1.** (Continued)

| Study ID (country) | Sample size and participant characteristics | Healthcare workers experiencing long-term symptoms, n/N (%) | Details of symptoms, n (%) unless specified |
|---|---|---|---|
| **Stepanek 2023** [39] **(Czech Republic)** | Total n = 305 (181 cases and 124 controls)<br>Occupation: NR<br>Age, Mean (95%CI): PCS, 47.3 (45.9, 48.8); controls, 42.4 (40.5, 44.3)<br>Sex: PCS, female 86.2%, male 13.8%; controls, female 71.8%, male 28.2% | 181/305 (59.3) [≥12 weeks after acute infection] | • Mean number of PCS symptoms: 1.9 (median 2)<br>• Persisting tiredness or fatigue that interfered with daily life: 86 (47.5%)<br>• Shortness of breath: 69 (38.1%)<br>• Muscle, joint or body aches: 29 (16%)<br>• Loss of smell: 27 (14.9%)<br>• Headache: 27 (14.9%)<br>• Sleep disorder: 20 (11%)<br>• Loss of taste: 17 (9.4%)<br>• Cough: 16 (8.8%)<br>• Chest pain or pressure: 14 (7.7%)<br>• Hair loss or skin problems: 8 (4.4%)<br>• Depression or anxiety: 5 (2.8%)<br>• Palpitations: 4 (2.2%)<br>• Rash: 4 (2.2%)<br>• Visual impairment: 3 (1.7%)<br>• 'Brain fog': 2 (1.1%)<br>• Following symptoms reported by one participant (0.6%) each: runny nose, fever, pins-and-needles, diarrhoea, sweating and speech disorders |

Note. NR: not reported; UK: United Kingdom; IQR: interquartile range; y: years; PCS: post-COVID-19 syndrome; HCW: healthcare workers; SD: standard deviation

[a]Seropositive participants only;

[b]181 participants at days 31–60 and 148 participants at days 61–90 (positive PCR test); 581 participants at days 31–60 and 515 participants at days 61–90 (negative PCR test). Percentages shown are for participants with a positive PCR test and as proportions of daily recordings of the symptom

[c]Original publication does not report exact numbers of participants experiencing each symptom. Therefore, percentages are approximate.

[d]PCR evidence of infection

professionals. They felt their voice as a patient was not heard and their symptoms were not taken seriously, a finding from most lived experience LC studies.

The participants clearly described the difficulty of combining their dual role as healthcare workers and patients and some recognised the challenge their doctors faced in managing a novel condition but felt that the onus was on themselves to provide answers to their questions.

Evidence already exists in the literature on how healthcare workers are susceptible to physical and mental illness [40, 41]. It is, therefore, no surprise that the studies included in this systematic review reported that long-term symptoms following COVID-19 infection were common among healthcare workers. In the survey studies, physical symptoms were reported more frequently than psychological symptoms but having professional medical knowledge did not protect the healthcare workers from the uncertainty and consequent fear about the nature and course of their symptoms. Furthermore, working in the healthcare sector was not an advantage in finding appropriate care. The impact of the problems experienced by people who experienced LC and the need to be listened to and supported by their doctors has been documented in the literature [42]. A systematic review assessing the barriers health professionals experience in accessing healthcare has highlighted important similarities between them and the general population [43].

Healthcare workers and especially doctors tend to consider their professional identity their core identity, which is often associated with a strong sense of power and the belief to be 'invincible' [44]. It is, therefore, challenging for their medical self to recognise their own illness and vulnerability. The pre-COVID literature already shows that doctors who have been away from work because of illness tend to internalise the perceived negative response of colleagues and their families to their problems, consider themselves as failures, and express self-stigmatisation

**Table 2. Data related to the working life, personal life and healthcare resource use of participants with Long-COVID symptoms.**

| Study ID | Details |
|---|---|
| **Working life** | |
| **Gaber 2021** [13] | • 3/138 (2.2%) had sick leave following the initial leave during the acute phase of the infection |
| **Havervall 2021** [14] | • 8% of seropositive participants reported that their long-term symptoms moderately to markedly disrupted their work life, compared with 4% of seronegative participants (RR, 1.8 [95% CI, 1.2–2.9]) |
| **Martinez 2021** [15] | • 191/260 (73.5%) had symptoms lasting < 90 days and reported 1801 cumulative missed workdays (median 10, IQR 7–11)<br>• 37/260 (14.2%) reported persisting symptoms over 365 days and 106 cumulative missed workdays (median 21, IQR 18–21)<br>• 32/260 (12.3%) with symptom duration of 365 days or less reported 303 cumulative missed workdays (median 10, IQR 5–10) |
| **Kaplan 2022** [26] | • 40/121 (33.1%) reported fatigue for >3 weeks; 29/40 (72.5%) could carry out their daily work |
| **Mendola 2022** [28] | • Perceived work ability at COVID-19 recovery, median (IQR): 8 (5.25–10)<br>• Perceived work ability at 18 months after infection, median (IQR): 9 (8–10)<br>• Fitness to work before COVID-19, n (%):<br>  ○ Fully fit to work: 39 (69.6)<br>  ○ Fit with restrictions: 17 (31.4)<br>• Fitness to work at time of return to work after COVID-19, n (%):<br>  ○ Fully fit to work: 18 (39.1%)<br>  ○ Temporarily not fit: 1 (2.2%)<br>  ○ Fit with restriction: 27 (58.7%) |
| **Mohr 2022** [30] | • At 6 weeks after symptom onset, 1.7% of HCWs had not returned to work<br>• HCWs who reported COVID-like symptoms on return to work were more likely than those without to report COVID-like symptoms at 6 weeks (84.7% vs 50.9%; RR 1.36, 95% CI 1.11 to 1.67).<br>• Vaccinated HCW returned to work a median 2.0 days (95% CI 1.0 to 3.0) sooner than unvaccinated HCW (adjusted HR 1.37, 95% CI 1.04 to 1.79) |
| **Personal life** | |
| **Havervall 2021** [14] | • 15% of seropositive participants reported their long-term symptoms moderately to markedly disrupted their social life, compared with 6% of seronegative participants (RR, 2.5 [95% CI, 1.8–3.6])<br>• 12% of seropositive participants reported that their long-term symptoms moderately to markedly disrupted their home life, compared with 5% of seronegative participants (RR, 2.3 [95% CI, 1.6–3.4]) |
| **Pereira 2021** [18] | • 6/38 (16%) reported that they were no longer able to participate in a sport or recreational activity because of their ongoing symptoms |
| **Rao 2021** [19] | Post-recovery, 71% were leading a healthier lifestyle, with 65% practising some form of physical exercise or yoga, and 47% taking health supplements, mostly multivitamins and Vitamin C:<br>  • Multivitamins: 56.63%<br>  • Vitamin C: 62.65%<br>  • Zinc supplements: 26.51%<br>  • Protein: 12.05%<br>  • Ayurvedic/homeopathic: 8.43% |
| **Kaplan 2022** [26] | • 40/121 (33.1%) reported fatigue for >3 weeks; 5/40 (12.5%) spent <50% of the day in bed; 3/40 (7.5%) spent >50% of the day in bed; no participants reported total bed confinement<br>• 91/121 (76%) reported no problems with mobility; 25/121 (20.7%) reported slight problems<br>• 114/121 (94.2%) reported no problems with self-care; 7/121 (5.8%) reported slight problems<br>• 103/121 (85.1%) reported no problems with usual activities; 14/121 (11.6%) reported slight problems; 4/121 (3.3%) reported moderate problems |
| **Healthcare resource use** | |
| **Gaber 2021** [13] | • 16/138 (11.6%) consulted their GP about their symptoms |

RR, risk ratio; CI, confidence interval; GP, general practitioner

views, which represent major barriers to returning to work [45]. Continuing to improve the training that medical students receive and remodelling of the general perception that 'doctors

**Table 3. Risk of bias of included studies, as assessed by the quality of reporting tool.**

| Study ID | Research question/study design | Participant selection & recruitment | Data collection | Analysis methods |
|---|---|---|---|---|
| **Brandt 2021 [12]** | No/no | Yes | Yes | Unclear |
| **Gaber 2021 [13]** | No/no | No | No | No |
| **Havervall 2021 [14]** | Yes/yes | Yes | Yes | Yes |
| **Ladds 2021 [10]** | Yes/yes | Yes | Yes | Yes |
| **Martinez 2021 [15]** | Yes/no | Yes | Yes | Yes |
| **Mattioli 2021 [16]** | Yes/yes | Yes | Yes | Yes |
| **Nielsen 2021 [17]** | Yes/no | Yes | Yes | Yes |
| **Pereira 2021 [18]** | Yes/no | Yes | No | Yes |
| **Rao 2021 [19]** | Yes/no | No | Yes | Yes |
| **Sultana 2021 [20]** | Yes/no | Yes | Yes | Yes |
| **Tawfik 2021 [21]** | Yes/no | Yes | No | Yes |
| **Taylor 2021 [11]** | Yes/yes | Yes | Yes | Yes |
| **Tempany 2021 [22]** | Yes/no | Yes | Yes | Yes |
| **Akova 2022 [23]** | No/no | Yes | Yes | Yes |
| **Carazo 2022 [24]** | Yes/no | Yes | Yes | Yes |
| **Kameyama 2022 [25]** | Yes/no | Yes | Yes | Yes |
| **Kaplan 2022 [26]** | Yes/no | Yes | Yes | Yes |
| **Kinge 2022 [27]** | Yes/no | Yes | Yes | Yes |
| **Mendola 2022 [28]** | Yes/no | Yes | Yes | Yes |
| **Mohr 2022 [29]** | Yes/no | Yes | Yes | Yes |
| **Nehme 2022 [30]** | Yes/no | Yes | Yes | Yes |
| **El Otmani 2022 [31]** | Yes/no | Yes | Yes | Unclear |
| **Pilmis 2022 [32]** | Yes/no | Yes | Yes | Yes |
| **Selvaskandan 2022 [33]** | Yes/no | Yes | Yes | Yes |
| **Senjam 2022 [34]** | Yes/no | Yes | Yes | Yes |
| **Strahm 2022 [36]** | Yes/no | Yes | Yes | Yes |
| **Uvais 2022 [37]** | Yes/no | Yes | Yes | Yes |
| **D'Avila 2023 [38]** | Yes/no | Yes | Yes | Yes |
| **Shukla 2022 [35]** | Yes/no | Yes | Yes | Yes |
| **Stepanek 2023 [39]** | Yes/no | Yes | Yes | Yes |

are invincible' may allow doctors to maintain their strong medical identity but be more accepting of their own limits [44].

## Strengths and limitations of the review

Extensive searches were conducted to identify relevant literature and two reviewers were involved in the selection of relevant studies and data extraction. Despite comprehensive searches, it is possible that relevant literature was not identified and it is likely that further relevant literature has since been published, given the fast-paced nature of research into the COVID-19 epidemic and its long-term sequelae. However, in the context of a rapid review, the methods used were robust and by current methodological standards. A potential limitation of our review is that we were not able to investigate associations between the effects of vaccination and LC symptoms, or the difference in LC symptoms between males and females, as they were not reported consistently by included studies. We recommend that future studies consider these potentially informative aspects. There was limited research on UK NHS workers, and the participants of these studies were largely doctors, white and from Western

populations. A more representative approach is needed to reflect the diverse occupations and ethnically varied nature of HCW.

## Conclusion

Having a medical background did not help healthcare professionals make sense of the wide range of debilitating and unpredictable LC symptoms. The dual role of being a patient and a doctor was particularly problematic and they felt dismissed and unheard by their doctors/clinicians. They reported a variety of persisting symptoms but low levels of sick leave and the need for multidisciplinary care was highlighted. There was little research on NHS workers and participants were mainly doctors, white and from Western populations.

## Supporting information

**S1 Appendix.**
(DOCX)

**S1 Table. Minimal dataset 1.**
(XLSX)

**S2 Table. Minimal dataset 2.**
(DOCX)

**S3 Table. Minimal dataset 3.**
(DOCX)

## Author Contributions

**Conceptualization:** Miriam Brazzelli, Nicola Torrance, Aileen Grant.

**Data curation:** Miriam Brazzelli, Paul Manson, Aileen Grant.

**Formal analysis:** Moira Cruickshank, Miriam Brazzelli.

**Funding acquisition:** Nicola Torrance, Aileen Grant.

**Methodology:** Miriam Brazzelli, Nicola Torrance.

**Project administration:** Moira Cruickshank.

**Software:** Paul Manson.

**Supervision:** Miriam Brazzelli, Nicola Torrance, Aileen Grant.

**Writing – original draft:** Moira Cruickshank.

**Writing – review & editing:** Miriam Brazzelli, Paul Manson, Nicola Torrance, Aileen Grant.

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
