## [Decision Letter · Decision Letter 0]

16 Oct 2023

PONE-D-23-14983What is the impact of long-term COVID-19 on workers in healthcare settings? A rapid systematic review of current evidencePLOS ONE

Dear Dr. % LAST_NAME%,

Thank you for submitting your manuscript to PLOS ONE. After careful consideration, we feel that it has merit but does not fully meet PLOS ONE’s publication criteria as it currently stands. Therefore, we invite you to submit a revised version of the manuscript that addresses the points raised during the review process.

We look forward to receiving your revised manuscript.

Kind regards,

Akaninyene Eseme Bernard Ubom, MBBS, MWACS, OMI Fellow

Academic Editor

PLOS ONE

“COVID Scottish Funding Council Research Funding and the School of Nursing, Midwifery and Paramedic Practice at Robert Gordon University.”

“The authors received no specific funding for this work”

Reviewers' comments:

Reviewer's Responses to Questions

**Comments to the Author**

1. Is the manuscript technically sound, and do the data support the conclusions?

Reviewer #1: Yes

Reviewer #2: Partly

Reviewer #3: Yes

2. Has the statistical analysis been performed appropriately and rigorously? 

Reviewer #1: Yes

Reviewer #2: No

Reviewer #3: Yes

3. Have the authors made all data underlying the findings in their manuscript fully available?

Reviewer #1: Yes

Reviewer #2: No

Reviewer #3: Yes

4. Is the manuscript presented in an intelligible fashion and written in standard English?

Reviewer #1: Yes

Reviewer #2: No

Reviewer #3: Yes

5. Review Comments to the Author

Reviewer #1: 1. ABSTRACT: The abstract should be structured into Background, Objective, Methods, Results and Conclusion.

*Include a brief background for the study

*Discussion is usually not part of abstract

*The result section of the abstract should be shortened /summarized to highlight only the major and important findings.

2. Line 43-- “Rationale” should be deleted since the study has been justified in the latter part of the introduction

3. Line 91—“Objectives” should be deleted, lines 92-94 should be moved to the introduction as an aim of the study

4. Tables 1 and 2 appear quite complex and ambiguous. I suggest these 2 tables should be simplified and summarized to highlight and showcase the important and pertinent findings of the previous works done. Information like methods of data analysis and study objectives may be left out while information on participants’ characteristics and sample size should be summarized. What is more important is the summary of main findings.

5. Line 403 “Results of quantitative studies” --- This seems like a repetition of table 2. Kindly summarise or state what is not obvious from table 2

6. Table 4 seems more like a repetition of table 2.

7. Line 474 “Summary of findings” should be removed

8. Was there any difference in Long covid symptoms between males and females?

9. There should be a conclusion paragraph in the discussion which should summarise the findings of this study.

Reviewer #2: 11 -07-2023

From

Reviewer

To

Editor

PLOS ONE

Dear authors

Re: Reviewers’ comments on manuscript enlisted “What is the impact of long-term COVID-19 on workers in healthcare settings? A rapid systematic review of current evidence”. Manuscript ID - PONE-D-23-14983

Thank you for this invitation to review this manuscript. The authors should attend to the point-to-point raised queries. It is my utmost pleasure to review this manuscript. Please find enclosed my review comments.

1. Please was this review registered? Please quote the registration number.

2. ‘Discussion’ is NOT classically part of abstract. The abstract should highlight aim, methods, results, limitations and strength and conclusions from the work.

3. Could this be a limitation? Your work did not gather information that relates Long Covid with covid vaccination. Could those symptoms follow effect of vaccination or Long Covid?

4. Authors should please include the keywords used with extensions in their search strategy. Please revisit appendix 1 for a clearer search strategy and keyword representation.

5. Please what tool / software was used for data screening, and extraction? Please indicate.

6. “There was no restriction on language or study type at the search stage. Results were limited to those published from December 2019. Searches were all carried out in November 2021 and updated in December 2022.” Please did you exclude 2020 period in this study? See Line 135, 136, 137.

7. What tool was used in the assessment of list of Bias? Line 236

8. I suggest that the discussion section be shortened by half as this segment appear verbose.

9. A sub-analysis of long COVID versus covid vaccination was not considered in this study. Can this be a Limitation? This study was limited to hospital only, could this be a limitation?

10. Someone proficient in English to go through the whole manuscript correcting all syntax and typographical errors e.g. “The medicolegal aspect is huge, and I think possibly certainly feels that way as a GP and it’s scary to not be able to recognize potentially where you have deficits because if you can’t recognize them then that’s an UNKNOWN UNKNOWN in what can you do with that. And just the sort of fast-paced nature of GP and as…” See line 267. Repeated word highlighted.

11. I am not comfortable with the placement of tables in between text. International manuscript standards prefer tables and figures after the reference lists.

12. Authors should write in such a manner as to avoid plagiarism

Reviewer #3: Basically the writing is very good, but should be given a year limit, wheater the last 3 years so that the data is more relevant. Though there are grammatical and editorial problem that need correction in ur papaer.

6. PLOS authors have the option to publish the peer review history of their article (what does this mean?). If published, this will include your full peer review and any attached files.

Reviewer #1: **Yes: **Michael Sylvester Archibong

Reviewer #2: **Yes: **Dr. IGBODIKE Emeka Philip MBBS, FWACS, FMCOG

Reviewer #3: No

---

## [Author Response · Author response to Decision Letter 0]

11 Dec 2023

We have now attached a cover letter addressing the points raised:

Dear Dr Chenette

Manuscript: What is the impact of long-term COVID-19 on workers in healthcare settings? A rapid systematic review of current evidence

Please find enclosed a revised version of the above evidence synthesis for consideration of publication in PLOS ONE. We have revised the manuscript thoroughly according to the points raised during the review process. We are grateful to the reviewers for their valuable points about our work and for the subsequent opportunity to clarify our funding statement. 

We believe our work, which has been conducted to high methodological standards and represents a comprehensive and objective summary of the current relevant literature, is of interest to scientists in medicine and health service-related fields.

Long COVID is a condition with relapsing and remitting symptoms, of which there are over 200 symptoms associated. The Long COVID population is dynamic with reinfection of COVID-19 exacerbating symptoms and increasing the risk of developing Long COVID. Office of National Statistics data from November 2022 shows the prevalence of Long COVID is higher in healthcare workers (3.78%) than the general population (2.67%). As the Long COVID literature grows, systematic reviews are emerging reporting prevalence, symptoms, summarising major findings, and lived experience; 

however, despite the higher prevalence of Long COVID in healthcare workers, there is yet to be a systematic review published on the impact in this population. 

The findings of our review will contribute to enhancing current knowledge of the impact of Long COVID symptoms on the health and well-being of healthcare workers and on their use of health service resources. Management of long COVID symptoms is still challenging. Our manuscript addresses the challenges that healthcare workers with Long COVID are facing, especially in terms of the uncertainty of unexpected and unidentified symptoms, access to healthcare and dual patient/professional identity. We show that there are clearly specific issues relating to the COVID-19 pandemic and its sequelae, that merit consideration and attention.

For information, the University of Aberdeen has an open access agreement with PLOS.

We look forward to your decision.

Yours sincerely,

Dr Moira Cruickshank

mcruickshank@abdn.ac.uk

---

## [Editor Report · Decision Letter 1]

18 Dec 2023

PONE-D-23-14983R1What is the impact of long-term COVID-19 on workers in healthcare settings? A rapid systematic review of current evidencePLOS ONE

Dear Dr. Cruickshank,

Thank you for submitting your manuscript to PLOS ONE. After careful consideration, we feel that it has merit but does not fully meet PLOS ONE’s publication criteria as it currently stands. Therefore, we invite you to submit a revised version of the manuscript that addresses the points raised during the review process.

We look forward to receiving your revised manuscript.

Kind regards,

Akaninyene Eseme Bernard Ubom, MBBS, MWACS, OMI Fellow

Academic Editor

PLOS ONE

Additional Editor Comments:

Even though the authors have satisfactorily addressed most of the reviewers’ queries and comments, this work remains too verbose and lengthy.

-The Introduction should be abbreviated to 300-400 words maximum

-The methodology should be made more concise and precise leaving out unnecessary details.

-Results of qualitative studies: Reduce quotes to a maximum of 2 salient and relevant quotes under each theme

-Results of quantitative results: Listing every symptom and the number of studies where each was found is unnecessary. List the most important/most common symptoms and other symptoms can be mentioned collectively with appropriate references.

-The discussion should be abbreviated to 500-600 words maximum. The Discussion should explain the study findings with relevant literature references/citations and also compare with similar or previous studies.

---

## [Author Response · Author response to Decision Letter 1]

2 Feb 2024

The Introduction should be abbreviated to 300-400 words maximum: The Introduction has been shortened accordingly

The methodology should be made more concise and precise leaving out unnecessary details: The Methods section has been revised to focus on pertinent details

Results of qualitative studies: Reduce quotes to a maximum of 2 salient and relevant quotes under each theme: A maximum of two quotes have been provided under each theme

Results of quantitative results: Listing every symptom and the number of studies where each was found is unnecessary. List the most important/most common symptoms and other symptoms can be mentioned collectively with appropriate references: These results have been revised accordingly

The discussion should be abbreviated to 500-600 words maximum. The Discussion should explain the study findings with relevant literature references/citations and also compare with similar or previous studies: The Discussion has been shortened and accordingly

---

## [Editor Report · Decision Letter 2]

15 Feb 2024

What is the impact of long-term COVID-19 on workers in healthcare settings? A rapid systematic review of current evidence

PONE-D-23-14983R2

Dear Dr. Cruickshank,

We’re pleased to inform you that your manuscript has been judged scientifically suitable for publication and will be formally accepted for publication once it meets all outstanding technical requirements.

Kind regards,

Akaninyene Eseme Bernard Ubom, MBBS, MWACS, OMI Fellow, OWE Fellow

Academic Editor

PLOS ONE

Additional Editor Comments (optional):

Authors have satisfactorily addressed all queries and suggestions.
---

## [Editor Report · Acceptance letter]

22 Feb 2024

PONE-D-23-14983R2 

PLOS ONE

Dear Dr. Cruickshank, 

I'm pleased to inform you that your manuscript has been deemed suitable for publication in PLOS ONE. Congratulations! Your manuscript is now being handed over to our production team.

Kind regards, 

on behalf of

Dr. Akaninyene Eseme Bernard Ubom 

Academic Editor

PLOS ONE